# Manufacturing Aluminum/Multiwalled Carbon Nanotube Composites via Laser Powder Bed Fusion

**DOI:** 10.3390/ma13183927

**Published:** 2020-09-05

**Authors:** Eo Ryeong Lee, Se Eun Shin, Naoki Takata, Makoto Kobashi, Masaki Kato

**Affiliations:** 1Department of Materials Science and Metallurgical Engineering, Sunchon National University, Suncheon, Jeollanam-do 57922, Korea; cksh23@s.scnu.ac.kr; 2Department of Materials Process Engineering, Graduate School of Engineering, Nagoya University, Furo-cho, Chikusa-ku, Nagoya 464-8603, Japan; takata.naoki@material.nagoya-u.ac.jp (N.T.); kobashi.makoto@material.nagoya-u.ac.jp (M.K.); 3Aichi Center for Industry and Science Technology, 1267-1 Akiai, Yakusa-cho, Toyota 470-0356, Japan; ntakata@mtl.titech.ac.jp

**Keywords:** metal–matrix composite, ball milling, laser powder bed fusion, microstructures, nanoindentation

## Abstract

This study provides a novel approach to fabricating Al/C composites using laser powder bed fusion (LPBF) for a wide range of structural applications utilizing Al-matrix composites in additive manufacturing. We investigated the effects of LPBF on the fabrication of aluminum/multiwalled carbon nanotube (Al/MWCNT) composites under 25 different conditions, using varying laser power levels and scan speeds. The microstructures and mechanical properties of the specimens, such as elastic modulus and nanohardness, were analyzed, and trends were identified. We observed favorable sintering behavior under laser conditions with low energy density, which verified the suitability of Al/MWCNT composites for a fabrication process using LPBF. The size and number of pores increased in specimens produced under high energy density conditions, suggesting that they are more influenced by laser power than scan speed. Similarly, the elastic modulus of a specimen was also more affected by laser power than scan speed. In contrast, scan speed had a greater influence on the final nanohardness. Depending on the laser power used, we observed a difference in the crystallographic orientation of the specimens by a laser power during LPBF. When energy density is high, texture development of all samples tended to be more pronounced.

## 1. Introduction

Laser powder bed fusion (LPBF) is a promising additive manufacturing technology that enables three-dimensional (3D) shape flexibility in feedstock with various metallic powders at a fast production rate [1]. LPBF minimized material waste, which also helps reduce production costs [2]. In the LPBF process, a computer-controlled laser selectively melts the powder bed, which is iterated layer by layer until the final product [3]. Owing to the advantages of LPBF, many studies have tried to analyze and report on commercial alloy powders, such as stainless steels (e.g., AISI316L [4]), titanium (e.g., Ti-6Al-4V [5]), aluminum (e.g., AlSi10Mg [6]), and nickel-based superalloys (e.g., Inconel [7]). However, the porosity has still a challenging issue in LPBF, e.g., Li et al. [4] reported that a yield strength (YS) decreased as porosity increased from 5% to 35% in AISI316L. Further, several studies investigated the effect of heat treatment on the mechanical properties with microstructural stability, such as enhanced elongation of selective laser melted AlSi10Mg from 1.4% to 3.9% due to homogeneous microstructure, as reported by Aboulkhair et al. [6] or improved strength owing to precipitates in the maraging steels, as described by Jagle et al. [7].

Although the microstructural stability of these materials was still insufficient for industrial applications, these alloys were successfully used in LPBF to fabricate structures with improved mechanical properties. For example, Carlton et al. [8] introduced the laser powder bed fusion-processed (LPBFed) AISI316L showed an increase of 357% in YS that was more than twice as high as that of wrought alloy, which revealed that the LPBF process improves the mechanical properties of the material. Xu et al. [9] reported that LPBFed Ti-6Al-4V alloys exhibited above 1 GPa in YS and 11% in elongation, which is much higher than that of forged Ti alloys, and that many industrial parts have been successfully manufactured using Ti-6Al-4V (Rafi et al. [10]). LPBFed AlSi10Mg alloys have been studied because of their excellent flow rate and narrow solidification temperature range in a molten state. The study by Schmidtke [11] presented the LPBFed AlSi10Mg showing a YS of 275 MPa with 8% elongation, which is slightly better than that of wrought alloy (YS 140 MPa with 1%). 

Further studies analyzed the relationship between the heat treatment effect and mechanical properties (e.g., tensile strength and nanohardness) of LPBFed AlSi10Mg among other mechanical properties such as the work by Takata et al. [12]. Although Al alloys are widely used in many industries, only a few are currently available for LPBF processes (e.g., AlSi10Mg and AlSi12) due to their high reflectance and thermal conductivities. Low laser absorption weakens the melt efficiency of the melt pool, increasing porosity and consequently, deteriorating the mechanical properties of the product [6]. 

On the other hand, lightweight and high-strength multifunctional materials can be tailor-made using powder metallurgy and metal–matrix composites (MMCs) for specialized applications. For instance, Zhao et al. [13] analyzed Fe-matrix reinforced with titanium carbides (TiC) as a function of TiC contents. The composite was successfully processed, although the high-temperature environment of LPBF caused the dissolution of C and Ti in the Fe matrix and deteriorated the mechanical properties of the fabricated object. Additionally, Attar et al. [14] presented that the Ti-matrix composites have shown almost full density (>99.5%) and high strength (~1 GPa) and hardness (~400 Hv).

Recently, Jiang et al. [15] reported on the difficulty of manufacturing due to splash phenomenon or microcracks when manufacturing the Al-matrix composites using the LPBF process. After the laser process, a relative density of slightly over 95% was observed in AlSi10Mg-matrix composites reinforced with multiwall carbon nanotubes (MWCNTs). Aboulkhair et al. [16] reported that the LPBF processed microstructures of aluminum/multiwalled carbon nanotube (Al/MWCNT) composites involved the formation of semicrystalline carbides (Al_4_C_3_) and partial regraphitization of CNTs with a nanohardness of 0.54 GPa. Jiang et al. [15] showed that LPBFed AlSi10Mg/MWCNT composites exhibited approximately 20% higher strength than the commercial alloy AlSi10Mg. The study, however, did not investigate the strengthening mechanism in detail.

Previous research on the fabrication of pure Al/MWCNT composites via LPBF have not been sufficiently conducted, because pure Al is difficult to process with lasers, leading to poor application of the LPBF process. The goal of this paper is to present and analyze the microstructures of LPBFed Al/MWCNT composite processed under 25 different conditions and to use the results as a guideline for creating Al-C material systems. In this study, we use the LPBF process to produce an Al composite powder with high laser absorption and excellent mechanical properties and containing MWCNTs. The experiment enables us to determine LPBF conditions, such as microstructural and mechanical properties, that are favorable for fabricating composite powder.

## 2. Materials and Methods

### 2.1. Powder Preparation

Al powder (99.5% purity and diameter less than 75 μm; sourced from Al Co., Ltd., Tokyo, Japan) and 1 vol.% MWCNT (20 nm in diameter and 5 μm in length; sourced from Applied Carbon Nano Co., Ltd., Pohang, Korea) were used as a matrix and a reinforcing material, respectively. High-energy ball milling was performed using an attrition mill filled with Al powder, MWCNT, stainless steel balls (~5 mm diameter) and 1% by weight stearic acid (CH_3_(CH_2_)_16_CO_2_H) in a stainless steel chamber (vertical tank, 2 L) for 24 h in a refined argon atmosphere at 500 RPM (Revolution per Minute). The ball to powder weight ratio set 15:1, and the chamber was agitated at room temperature, using a vertical shaft inserted horizontal impellers. The morphology of Al powder, MWCNT powder, and ball-milled Al/MWCNT composite powder was analyzed using a scanning electron microscope (SEM) (Figure 1). The particle size of the composite powder was 16 ± 6 µm, which was applied by sieving for the LPBF process with the powder bed layer thickness set to 30 µm.

### 2.2. LPBF Processing

Ball-milled Al/MWCNT composite powder was used as feedstock to fabricate cuboidal samples (15 mm × 15 mm × 1 mm) with LPBF under 25 conditions by altering the laser power *P_L_* (102, 128, 153, 179, and 204 W) and scan speeds vs. (1.4, 1.2, 1.0, 0.8, and 0.6 m/s). LPBF treatment was performed at room temperature using a ProX 200 (3D Systems, Rock Hill, SC, USA) additive manufacturing system equipped with a Yb-doped fiber laser. High-purity Ar gas was added to the chamber to prevent oxidation of the sample prepared during the LPBF process. To avoid the interaction of used composite powder with the base Al alloy plate, we applied a thin support structure layer with a thickness of approximately 1 mm to the surface of the base plate (as presented in Figure 2a) before creating cuboidal samples. Specimens were separated from the base Al plate using wire cutting. In this experiment, a hexagonal grid laser scanning pattern was applied. The longest diagonals of the regular hexagons were set to 10 mm, and the laser scanning track was rotated 90° between two successive layered powder layers. The study of Takata et al. [17] has provided a detailed description of the scanning pattern. The hatch distance (hs) was 100 μm and the layer thickness (ds) was 30 μm. This resulted in 25 different linear laser energy densities (E_v_)—from 24.29 to 113.33 J/mm^3^—as defined by:E_v_ = P_L_/(v_s_ ∙ h_s_ ∙ d_s_),(1)

The laser densities were used to evaluate the effect of energy input per unit length in relation to layer results.

### 2.3. Analysis

Powder morphology was analyzed using field-emission scanning electron microscopy (FE-SEM; JSM-7100F, JEOL, Tokyo, Japan). The microstructure of the bulk Al/MWCNTs composite was examined using optical microscopy (OM; BX53MRF, Olympus, Tokyo, Japan), a SEM equipped with an electron backscatter diffractometer (EBSD; Hikari, EDAX-TSL, Tokyo, Japan). The accelerating voltage used for analysis was 20 kV, probe current was 12 nA, working distance was 15 mm, the step size was 0.02 μm, and the minimum confidence index was 0.1. Tomography was carried out using X-ray micro-computed tomography (micro-CT; Skyscan 1272, Bruker, Kontichm, Belgium). The X-ray system was equipped with a microfocus X-ray tube with a focal spot of 5 μm and produced a conical beam detected by a 12-bit cooling X-ray CCD camera connected by optical fiber to a 0.5 mm scintillator. The voxel (3D pixel) size did not exceed 5 μm and the reconstruction of 3D images was performed using the commercial software InstaRecon^®^ and multithreaded CPU/GPU 3D reconstruction.

To observe the nanoscale microstructures of the specimen, a high-resolution transmission electron microscopy (HRTEM; JEOL 2000, JEOL, Tokyo, Japan) was used, employing focused ion beam (FIB) milling techniques (FEI Helios 600 Dual Beam, Thermo Fisher, Hillsboro, USA). The crystal structure of the samples was analyzed using X-ray diffraction (XRD; Bruker, D8 discover, Kontich, Belgium) with a Cu Kα radiation source (λ = 1.5405Å) and an uninterrupted scan modem with a scan rate of 1°/min and a range of 2θ (20°–90°). A nanoindentation test was handled using a diamond Berkovich indenter (TI 950 Triboindenter, Hysitron, Eden Prairie, MN, USA), where the maximum indentation load was 50 mN, the maximum indentation depth was 25 nm, and the loading/unloading rate was 5 mN/min.

## 3. Results and Discussion

### 3.1. Fabricated and Morphologies of the LPBFed As-Deposited Specimens

Figure 2 shows the 25 LPBFed Al/MWCNT composite specimens fabricated with different scan speeds and laser powers. When we used high laser powers (e.g., 179 and 204 W), the surface of the specimens was cut off, resulting in poor outcomes. The seven poor types of specimens were not evaluated further. We grouped the remaining 18 specimens as follows: (1) L1, L2, and L3 groups created under laser powers of 102, 128, and 153 W, but with different scan speeds ranging from 1.4 to 0.6 m/s and (2) groups S1, S2, and S3 created under the same scan speeds (1.4, 1.2, and 1.0 m/s), but different laser powers (ranging from 102 to 204 W), respectively.

Figure 3 shows the overall trends in the specimens through OM and SEM images where specimens were grouped according to laser power and scan speed, excluding the incompletely built specimens. In Figure 3a, The Al/MWCNT composite was successfully prepared through the LPBF process, but black spots were observed on the surface of the specimen, which was confirmed to be pores. In group L1, the surface porosity of the specimens was less than 4.9%, which is lower than that of the other groups. The L1 specimen created with a scan speed of 1.4 m/s had a porosity of 4.0%, the lowest among all fabricated specimens. In group L2, the specimen created with a scan speed of 1.4 m/s showed a relatively low porosity of 11.0%. The L2 specimens fabricated with scan speeds of 0.6 and 0.8 m/s had the highest porosity (>17.5%) and showed the highest value among the L2 group. Results indicate that the effect of laser power on the surface porosity is insignificant when the energy density is below a certain level such as 40 J/mm^3^. However, porosity tended to rise in groups S1 and S2 as laser power was increased while employing constant scan speeds. In group S3, the porosity of the specimens increased for those fabricated at 102 to 153 W but then decreased for the specimen fabricated at a laser power of 179 W. Specimens fabricated with a laser power that was higher than 179 W (E_v_ = 51 J/mm^3^) were destroyed due to overheating of the surface. The one fabricated at 179 W could be used for the LPBF of Al/MWCNT composite powder, which is considered to be close to the limit of the energy densities that can be used for LPBFed Al/MWCNT composite powders. Therefore, when preparing the sample, the powder was perceived to have completely melted because the high energy may have slightly aided in reducing the porosity of the sample.

Similar results were observed in the SEM images (Figure 3b). Group L1 exhibited the most reliable surface, where surface porosity was lowest for the specimen fabricated at a scan speed of 1.4 m/s. However, in group L3, porosity increased with scan speed and, unlike the OM analysis, the porosity of specimens fabricated with scan speeds of 1.2 and 1.4 m/s showed the highest values: 12.2% and 16.4%, respectively. In addition, the results for groups S1, S2, and S3 confirmed that porosity increases with rising laser power. In the case of group S1, the porosity of the specimens radically increased when laser power exceeded 128 W (E_v_ = 30.48 J/mm^3^). Hence, in the low energy density range, the effect of laser power on porosity seems to be greater than that of scan speed.

### 3.2. Microstructural Evaluation of the LPBFed Specimens

The X-ray diffraction profiles of the ball-milled composite powder and LPBFed Al/MWCNTs composite specimens are shown in Figure 4. As a result of XRD analysis, aluminum (111), (200), (220), (311) peaks were detected in all samples of composite powders and composites. There were also specimens in which aluminum oxide (Al_2_O_3_) peaks and aluminum carbide (Al_4_C_3_) peaks were detected. The peak of MWCNT was not detected in all the XRD graphs including the ball-milled powder, because some of the carbon in MWCNT is generated to carbides in the Al base during the LPBF process involving high heat, and the residual MWCNT content is low. Also, it can be assumed that as the MWCNT was well inserted into the aluminum powder during high-energy ball milling, the amount of MWCNT remaining on the surface was small. Or it is possible that some of the MWCNTs are damaged due to excessive energy during milling. However, in the case of the aluminum composite with low concentration of MWCNT, the phenomenon that the peak of MWCNT was not well detected occurred in other documents than this study [18].

In order to confirm the trend in the XRD profiles regarding laser power, Figure 4b–d show groups L1, L2, and L3 with laser powers of 102, 128, and 153 W, respectively. As shown in Figure 4b, L1 samples with scan speeds of 1.4 and 1.0 m/s were provided with a similar form of ball-milled composite powder and resulted in a stronger intensity of the (111) peak compared to other groups. We hypothesized that the texture of the specimen changes as the energy density increases, but this pattern was not found in group S1, which had a lower energy density, indicating that the trend in the XRD profiles was not based on energy density. In addition, there is only a minimal change in the texture between the powder and the LPBFed samples when using weak laser power. The (222) peak was detected in all specimens of the L1 group, and (220) and (311) peaks were clearly detected when compared with other laser conditions. Figure 4c shows that in group L2, the intensity of the (200) peak was higher than that of the (111) peak, and this trend was also observed in group L3 (Figure 4d). XRD graphs of the samples fabricated with scan speeds of 1.4, 1.2, and 1.0 m/s are shown in Figure 4e–g, respectively. For samples fabricated with a relatively low laser power of 102 W, the intensity of the (111) peak was highest, whereas samples fabricated with higher laser powers (128–179 W) exhibited a higher intensity of the (200) peak compared to other samples. Our experiment shows that specimens fabricated with the same laser power will have similar aggregate structures.

### 3.3. Texture Evolution of LPBFed Specimens

Figure 5 includes inversed pole figure (IPF) maps and unit triangles of IPF, showing crystallographic textures along the building direction (BD) and scanning direction (SD) of the front and top view of LPBFed specimens in groups L1 and S2. The directions that are perpendicular and parallel to the building direction are termed SD_x_ or SD_y_ and BD. Figure 5a,b shows EBSD analyses of group L1 specimens fabricated with a laser power of 102 W and group S1 specimens fabricated with a scan speed of 1.4 m/s. IPF maps parallel to the *x*-axis and *y*-axis of BD and SD are also shown. The black parts of the EBSD image were pores and were not measured. 

Overall, we observed weak intensities of all texture components in both groups L1 and S2. This phenomenon in LPBFed Al has been reported consistently in other research [16]. When energy density is high, texture development of all samples tended to be more pronounced. In the case of group L1, we observed a developed (001) orientation in the direction parallel to the *x*-axis scanning direction (SDx), whereas energy density increased in group S1. Thus, (001) orientation was also developed in the direction parallel to the *y*-axis scanning direction (SDy) as well as in the direction parallel to SDx. In the EBSD analysis of group S1, we had difficulties identifying similar trends to those previously observed in XRD graphs in Figure 4. There was no clear change in the texture development of specimens based on a specific orientation, which may have multiple causes, such as in groups L1 and S1, where grains exist in irregular and equiaxed shapes, and grain size grew with increasing energy density. This may be caused by grain coarsening at high temperatures. However, for group L1, with identical laser power, the grain size of the specimens with scan speeds of 1.4, 1.2, 1.0, 0.8, and 0.6 m/s were 25.8, 23.7, 25.1, 26.4, and 36.5 μm, respectively. This indicates that as energy density increased, grain size grew slowly. Furthermore, there was no special difference between the grain size and the ball-milled composite powder sifted below 30 um provided in the LPBF process. On the other hand, group S1 specimens showed radical changes in grain size. The grain size of specimens fabricated with laser powers of 102, 128, 153, and 179 W were 25.8, 33.0, 45.8, and 56.4 μm, respectively. Hence, grain coarsening during the LPBF process was more affected by laser power than by scan speed. In addition, compared to the specimen fabricated with a scan speed of 0.6 m/s in group L1 (E_v_ = 56.7 J/mm^3^) and the specimen fabricated with a laser power of 179 W in group S1 (E_v_ = 42.6 J/mm^3^), specimens with lower energy density exhibited coarser grains by as much as 64%. The energy density of the specimen with a scan speed of 0.6 m/s in group L1 was higher, but the grain size of the specimen with a laser power of 179 W in group S1 was larger. In the LPBF process, it is supported by speculation that the grain coarsening according to the laser conditions is more influenced by the laser power than the scan speed.

Thus, the experiment shows that specimens produced by the LPBF process exhibit inhomogeneous microstructures as a result of repeated locally occurring melting and solidification, resulting in high anisotropic mechanical properties, which affect microstructures such as the shape and size of grains and textures within the material (Park et al. [19]). However, as there was no apparent texture development of the specimens in groups L1 and S1 for the direction parallel to BD, the anisotropy property does not apply to the LPBF samples produced in this study.

As shown in Figure 5c, we performed an EBSD analysis of the front side of the sample, fabricated with a scan speed of 0.6 m/s in group L1 (the highest energy density of 56.7 J/mm^3^ in group L1 as shown in Figure 5a), to investigate the microstructures in the direction parallel to the specimen BD. We discovered that the shape of the grain was more elongated than the plane perpendicular to the BD, which uses a melted powder layer (n layer) and the newly provided powder layer (n + 1 layer) during the LPBF process that forms as a result of repeated melting and solidification. The grain size was 74.9 μm, which was approximately twice as large as the 36.5 μm specimen that had been LPBFed under a scan speed of 0.6 m/s in group L1.

We observed no relationship between grain size and energy density, but the grain size was found to be larger in the parallel side to BD than on the perpendicular side to BD in all specimens. The reason for this is that the perpendicular side consists of a single layer of powder particles, whereas the melted powder particles are already present in the lower layer (n − 1 layer) on the parallel side, unlike the melting and solidification with adjacent powders. Therefore, grain coarsening occurred more in the n layer than in the n − 1 layer, which means the newly formed layer (n layer) is more favorable for heat transfer.

### 3.4. High-Resolution Microstructures of LPBFed Specimens

Figure 6 shows nanoscale microstructures of selected LPBFed specimens using HRTEM. Figure 6a,b shows the group L3 specimen fabricated with a laser power of 153 W and a scan speed of 1.0 m/s (E_v_ = 51 J/mm^3^), whereas Figure 6c,d show the group L2 specimen fabricated with a laser power 128 W and a scan speed of 0.8 m/s (E_v_ = 53.3 J/mm^3^). We found agglomerated Al_4_C_3_ around the grain boundaries in the Al matrix, along with the resulting deformation bands. They interfere with the interfacial bonding between the matrix and the MWNCT, thus impeding effective load transfer. The formation of Al_4_C_3_ was due to the reaction of the Al matrix with carbon from MWCNTs at a temperature of 700 °C, which is above the melting point of Al (660 °C). Therefore, the repeated melting and solidification process, accompanied by high melt pool temperatures during the LPBF process, promoted the bonding reaction between Al and carbon. The resulting carbide is acicular, and is formed as carbon is precipitated from molten Al supersaturated with carbon content during cooling. Carbide generation is unavoidable in LPBF with high process temperature, but Al_4_C_3_ has been reported to weaken the interface strength between MWCNT and matrix Al, making load transfer difficult. Since Al_4_C_3_ and MWCNT coexist in one specimen, the complex effect of these two crystals on mechanical properties will be expressed.

### 3.5. Relationship between Porosity and Mechanical Properties

To evaluate the relationship between mechanical properties and porosity, we analyzed specimens with X-ray micro-CT and nanoindentation tests. Table 1 and Figure 7 summarize the results. We compared the two specimen groups with the lower energy density range (24.29–56.67 J/mm^3^ for L1 and 24.29–42.62 J/mm^3^ for S1). We found that the highest density of a specimen (LPBFed under the laser power of 102 W and the scan speed of 1.4 m/s) was calculated as 69.3% from the images of X-ray micro-CT analysis. We plotted the relative densities of groups L1 and S1 specimens in Figure 7c. In case of group L1, scan speed decreased from 1.4 to 0.6 m/s and density decreased from 69.3% to 61.7%. The relative density plot is based on the specimen fabricated with a laser power of 102 W and a scan speed of 1.4 m/s, and it shows the same trend as the microstructural observations. It can be concluded that scan speed affects not only the amount of the surface porosity but also the number of internal pores. Therefore, when it is desired to manufacture a sample having a higher density, it is better to increase the scanning speed under the same laser power condition.

Group S1, however, shows a very-low density change rate of less than 5%, which is different from the microstructural observation (in Figure 2) that was obtained from the samples fabricated with laser powers of 102 and 128 W (having a low porosity) and the sample fabricated with a laser power of 153 W (having a high porosity). Due to the nature of the additive manufacturing method, as internal pores are produced by stacking the next layer differently from surface pores, the difference in porosity between specimens is smaller than between surfaces because the melted powder fills the gap.

As can be seen in the 3D analysis in Figure 7, keyhole-shaped pores were present on the surface of the specimens, which were not observed in the microstructural analysis. We found these keyhole-like pores mainly in specimens with high energy densities, such as the specimen fabricated with a scan speed of 0.6 m/s in group L1 and the specimen fabricated with a laser power of 179 W in group S1, both of which exhibit high energy densities of 56.67 and 42.62 J/mm^3^, respectively. This phenomenon can be a contributing factor to an increase in surface roughness because the rise in energy density increases not only the amount of pores but also their size. The reason for keyhole-like pore occurrence is that the partly oxidized powders interfere with the stacking of the next layer.

In order to evaluate the mechanical properties of the specimens, elastic modulus and nanohardness were measured using a nanoindentation test. As shown in Table 1, the specimen fabricated with a laser power of 102 W in group S2 showed the lowest value of elastic modulus at 63.65 GPa, whereas the specimen fabricated with a laser power of 179 W in group S3 showed the highest value at 84.77 GPa. In the case of groups L1 and L2, the average value of the elastic modulus for the specimens fabricated with laser powers of 102 and 128 W was ~71 GPa, whereas the specimens fabricated with laser powers of 153 W and 179 had elastic modulus greater than 77 GPa—a much higher value. Therefore, at constant scan speed conditions, specimens fabricated with a higher output laser power exhibited a higher elastic modulus than that of specimens fabricated with lower laser powers.

In addition, the nanohardness values show that the lowest value (0.92 GPa) belonged to the specimen fabricated with a scan speed of 1.2 m/s in group L2, whereas the highest value (1.45 GPa) belonged to the specimen fabricated with a scan speed of 1.0 m/s in group L3. For the specimens of groups L1, L2, and L3, fabricated under the same laser power, the average nanohardness was 1.22 ± 0.02 GPa, whereas the average nanohardness of the specimen in group S1, with higher scan speeds, was 1.05 GPa. Additionally, at a lower scan speed of 1.0 or 0.8 m/s, the nanohardness values were relatively high (i.e., above 1.32 GPa). Therefore, the experiment confirmed that, keeping laser power the same, the nanohardness is higher for specimens prepared at lower scan speeds.

Table 2 summarizes the manufacturing processes and resulting properties of the recently developed commercial Al alloys and Al-matrix composites. Compared to the Al/MWCNT composites fabricated using LPBF in previous studies, all specimens produced by this study showed higher nanohardness values and up to 31% higher elastic modulus. Moreover, there was no significant difference in physical properties compared to the specimens fabricated using powder metallurgy. There was no significant difference from the powder metallurgy method, which can obtain a high density, and the specimen produced in this study had a relatively low density, but was not inferior in terms of physical properties. Therefore, if it is possible to manufacture a specimen having reached a high density by proceeding more in-depth in this study, it is expected that specimens with advantageous physical properties can be obtained compared to commercial materials currently being studied.

## 4. Conclusions

In this study, an Al-MWCNT composite powder was produced using mechanical ball milling. The powder was subsequently used as feedstock to fabricate cuboidal-shaped objects using LPBF under 25 different conditions. Our observations are summarized below:(1)We discovered a favorable sintering behavior under laser conditions with low energy density rather than high energy density, which verified the suitability of Al/MWCNT composites for fabrication using LPBF.(2)Overall, unsatisfactory density values were found in all LPBFed specimens. However, the size and number of pores increased in specimens that we produced under high energy density conditions, suggesting that they are more affected by laser power than scan speed. The porosity of the specimen was related to the elastic modulus and, therefore, exhibited a similar tendency. As porosity increased, the elastic modulus decreased.(3)Depending on the laser power used, there was a difference in the crystallographic orientation between the specimens. However, this was difficult to analyze because of the weak texture intensity in the overall specimens. The reason of anisotropy is not clearly revealed. We detected an advantage that can have similar properties in the three-dimensional direction owing to the anisotropic mechanical property.(4)In general, Al_4_C_3_ is known to have an unfavorable effect on mechanical properties, although the LPBFed specimens fabricated in this study showed a few Al_4_C_3_ phases and retained MWCNTs in the Al matrix due to the high temperatures involved in laser melting. They exhibited better mechanical properties than those of the LPBFed Al/MWCNT composites from previous studies.(5)Different variables dominated the mechanical properties of fabricated specimens. The laser power was more effect on the elastic modulus of a specimen than scan speed. Similarly, scan speed had a greater influence on the final nanohardness than laser power.

This study presents important research for the fabrication of Al/C composites using LPBF for a wide range of structural applications while utilizing Al-matrix composites in additive manufacturing. However, further studies are needed to obtain higher densities and improve mechanical properties of Al/C composites. Optimization of the LPBF process maximizes the density of fabricated specimens, but further research on heat treatment is necessary to refine the process. In addition, the mechanical properties of the produced material need to be examined more closely in order to build on this study and enable future material development.

## Figures and Tables

**Figure 1 materials-13-03927-f001:**
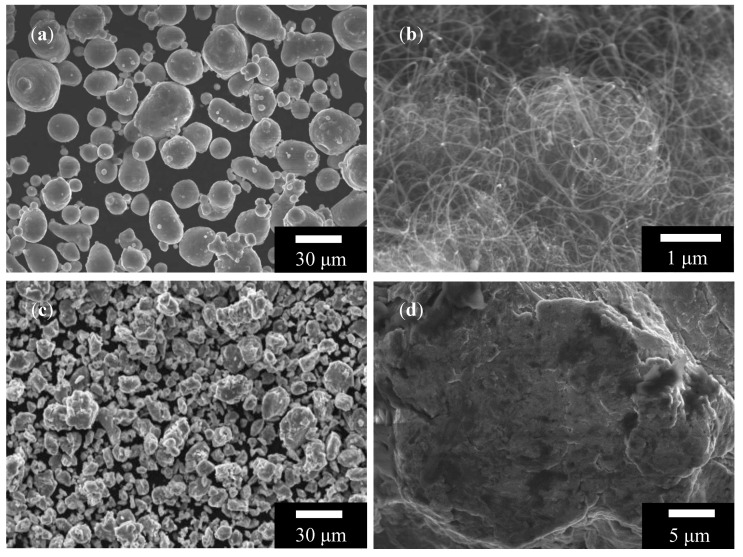
SEM images of (**a**) Al powder, (**b**) multiwalled carbon nanotube (MWCNT) powders, (**c**) ball-milled aluminum/multiwalled carbon nanotube (Al/MWCNT) composite powder, and (**d**) enlarged image of (**c**).

**Figure 2 materials-13-03927-f002:**
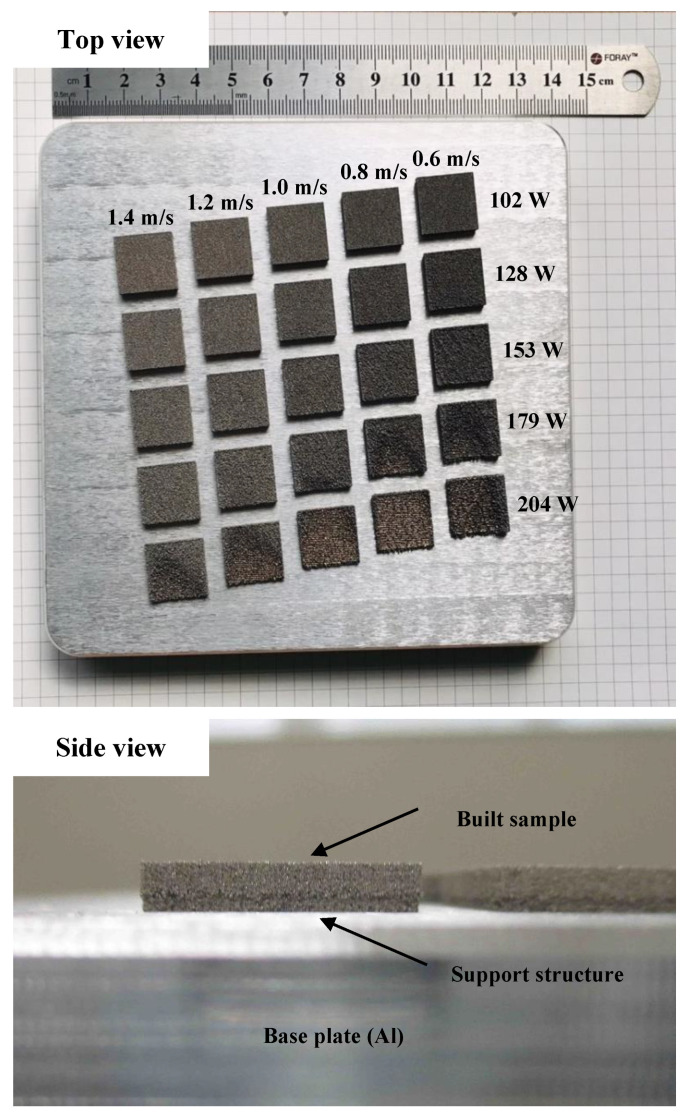
Physical appearance of the 25 cuboidal samples.

**Figure 3 materials-13-03927-f003:**
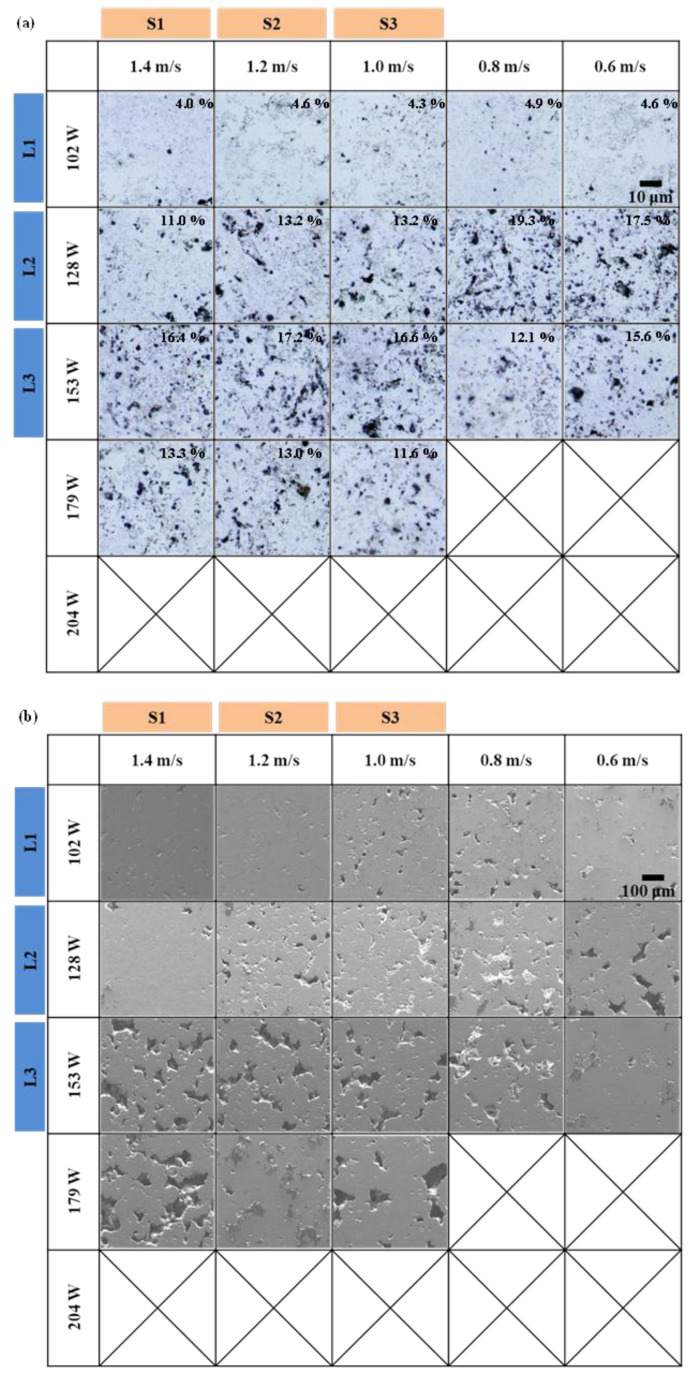
(**a**) Optical images and (**b**) SEM images of the laser powder bed fusion-processed (LPBFed) Al/MWCNTs composites, which were completely built (porosity is shown at the top of each image).

**Figure 4 materials-13-03927-f004:**
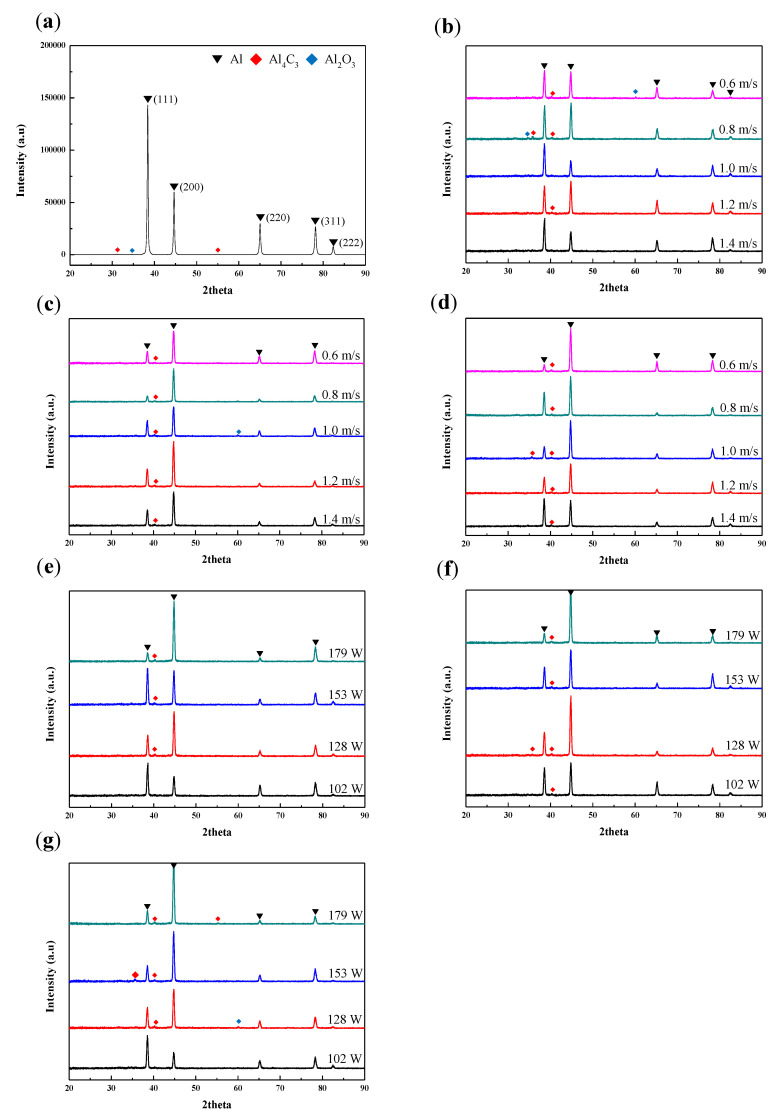
XRD patterns of the LPBFed Al/MWCNT specimens in each group: (**a**) ball-milled Al/MWCNT composite powder, (**b**) L1, (**c**) L2, (**d**) L3, (**e**) S1, (**f**) S2, and (**g**) S3.

**Figure 5 materials-13-03927-f005:**
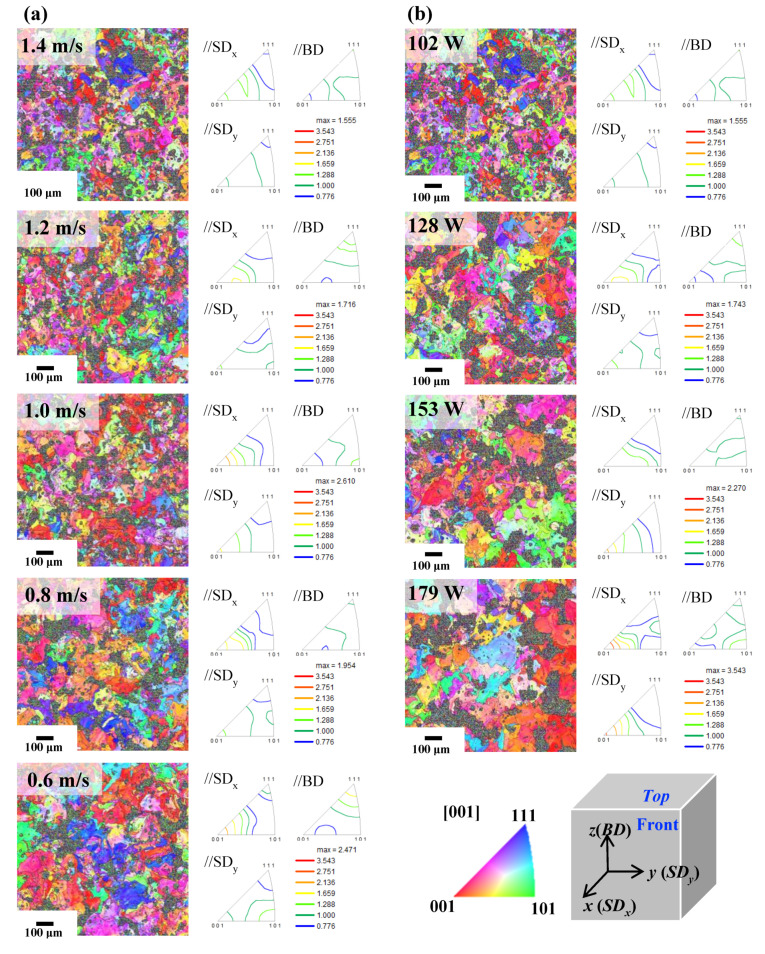
Electron backscatter diffractometer (EBSD) orientation color maps and pole figures of the LPBFed Al/MWCNTs composite specimens as a function of the laser power and scan speed; specimen with top view in groups (**a**) L1 and (**b**) S1 and front view in (**c**) group L1.

**Figure 6 materials-13-03927-f006:**
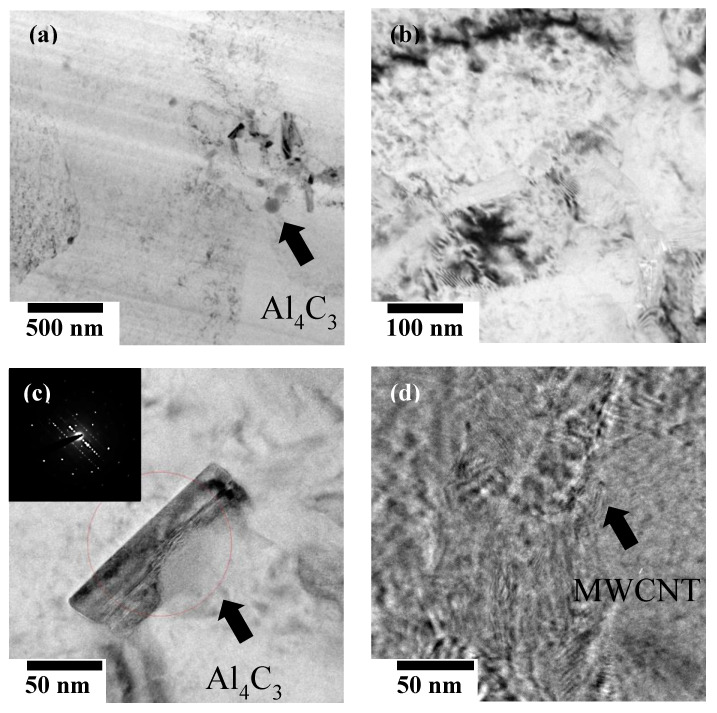
High-resolution transmission electron microscopy (HRTEM) images of the LPBFed Al/MWCNTs composite specimens with scan speeds of (**a**,**b**) 1.0 m/s in group L3 and (**c**,**d**) 0.8 m/s in group L2.

**Figure 7 materials-13-03927-f007:**
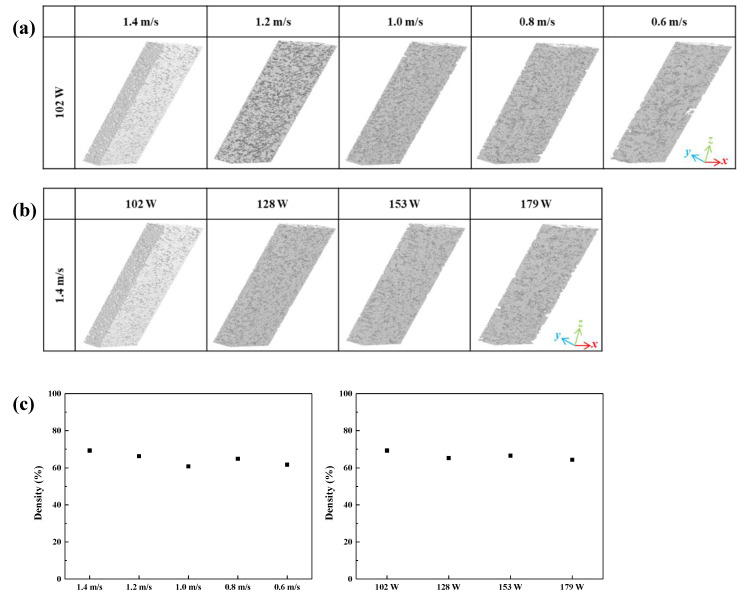
X-ray micro-computed tomography (micro-CT) images of the LPBFed Al/MWCNTs composite specimens in (**a**) group L1 and (**b**) group S1. (**c**) Relative densities of groups L1 and S1.

**Table 1 materials-13-03927-t001:** Elastic modulus and nanohardness of the laser powder bed fusion-processed (LPBFed) aluminum/multiwalled carbon nanotube (Al/MWCNT) composite specimens under different conditions.

Laser Power	Mechanical Properties	Scan Speed			
		1.4 m/s	1.2 m/s	1.0 m/s	0.8 m/s
102 W	Elastic modulus (GPa)	76.45 ± 1.17	63.65 ± 0.98	70.68 ± 1.33	75.41 ± 1.91
	Nanohardness (GPa)	1.27 ± 0.06	1.15 ± 0.02	1.15 ± 0.02	1.38 ± 0.03
128 W	Elastic modulus (GPa)	66.09 ± 2.01	69.60 ± 1.47	64.49 ± 2.23	84.19 ± 1.02
	Nanohardness (GPa)	1.13 ± 0.02	0.92 ± 0.03	1.33 ± 0.01	1.44 ± 0.01
153 W	Elastic modulus (GPa)	73.15 ± 1.36	79.77 ± 2.57	82.58 ± 1.95	73.02 ± 0.96
	Nanohardness (GPa)	1.14 ± 0.02	1.10 ± 0.02	1.45 ± 0.04	1.19 ± 0.03
179 W	Elastic modulus (GPa)	82.02 ± 1.85	75.75 ± 1.42	84.77 ± 2.28	67.23 ± 1.23
	Nanohardness (GPa)	1.30 ± 0.01	1.04 ± 0.04	1.40 ± 0.06	1.28 ± 0.02

**Table 2 materials-13-03927-t002:** Comparison of mechanical properties of Al/MWCNT composite specimens fabricated via laser powder bed fusion (LPBF) and powder metallurgy (NA: not provided).

Materials	Fabrication Method	Mechanical Properties		Reference
		Elastic Modulus (GPa)	Nanohardness (GPa)	
AlSi10Mg	SLM	100 ± 1.00	1.82 ± 0.01	[6]
AlSi5Ni	PM	NA	2.2 ± 0.35	[18]
Al/1MWCNT	SLM	63.65~84.77	0.92~1.45	Present study
Al/2MWCNT	PM	71.50	1.24	[15]
	SLM	64.7 ± 0.90	0.54 ± 0.01	

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
