# Peer review of "Manufacturing Aluminum/Multiwalled Carbon Nanotube Composites via Laser Powder Bed Fusion"

_materials, 2020, doi:10.3390/ma13183927_

Round 1

Reviewer 1 Report

The subject of the article and introduced research methods are quite interesting. The content of the article is logical and transparent. Generally, the article is quite well wrtitten. I have a few comments and questions  that need to be clarified:

1). Numerous minorrs for example: (page 2 line 90 and 91) "less than 75 m, ...and 5 m.( page 2 line 94)(CH3(CH2) 16CO2H) should be - (CH3(CH.....,. (page 5 line 165 and 169, 180, page 11 line 252, 253, 266)  should be - 40 J/m etc...., (page 7 line 191) should be Al2O3 and Al4C3.......

This kind of mistakes are visible in the all text.

2). In Fig. 2. is a) but were is b)? Correct the caption of the figure.

The magnification in Fig.2  is too small and the mactrostructure is invisible. Leaving the Fig 2. (page 5 line 147) without any comment introduce litle information into the text. 

3). Fig. 5. Not all microareas on the EBSD maps are analyzed (see black spots on EBSD maps). Please comment this situation. 

4). Fig. 6c. Electron diffraction should be solved. Moreover the diffraction spots should be well visible. 

5). Page 12 line 288 (Fig.6.) Where in Fig.6 are visible deformation bands. Please comment this situation.

6). In the text there is too little discussion of research. (Expect the data in Table 2). Especially, the discussion should be evident in science article, if not the article resemble the research report. 

Reviewer 2 Report

This is a very interesting research. A few comments:

  1. the paper requires extensive editing with language. There are several grammatical errors.
  2. Lines 90-91: "mu" symbol is missing when the authors specified the powder particle diameter.
  3. Line 94: the name should be (CH3(CH2) 16CO2H). Please use aproapriate subscript.
  4. Line 95: What is RPM?
  5. Line 144: Change 3.1 caption to "As-Deposited Specimens"
  6. Line 146: No (a) or (b) marked on Figure 2
  7. Figure 3: What is the point of saying that the authors have fabricated specimens 204W when no images are provided?
  8. What happened to 179W 0.8 m/s and 0.6 m/s?
  9. Line 191 and entire text: Al2O3 and Al4C3 - Please use appropriate subscripts.
  10. Figure 4: Please re-plot and increase the font sizes of the axes and captions. 
  11. Figure 5 quality is very poor. The resolution is not publication worthy.
  12. All over the paper the authors have used Ev, it should be Ev. Aslo, J/mm3 should be J/mm3
  13. Table 1: 204W properties are missing
  14. Figure 7c - Please improve the figure quality. The axes are barely visible. 

Round 2

Reviewer 1 Report

I accept the authors answers.